# Effect of Vagus Nerve Stimulation on the GASH/Sal Audiogenic-Seizure-Prone Hamster

**DOI:** 10.3390/ijms25010091

**Published:** 2023-12-20

**Authors:** Jaime Gonçalves-Sánchez, Consuelo Sancho, Dolores E. López, Orlando Castellano, Begoña García-Cenador, Gabriel Servilha-Menezes, Juan M. Corchado, Norberto García-Cairasco, Jesús M. Gonçalves-Estella

**Affiliations:** 1Department of Cellular Biology and Pathology, School of Medicine, University of Salamanca, 37007 Salamanca, Spain; lopezde@usal.es (D.E.L.); orlandoc@usal.es (O.C.); 2Institute for Biomedical Research of Salamanca (IBSAL), 37007 Salamanca, Spain; sanchoc@usal.es (C.S.); mbgc@usal.es (B.G.-C.); corchado@usal.es (J.M.C.); jgoncalves@usal.es (J.M.G.-E.); 3Institute of Neuroscience of Castilla y León, 37007 Salamanca, Spain; 4Department of Physiology and Pharmacology, School of Medicine, University of Salamanca, 37007 Salamanca, Spain; 5Department of Surgery, School of Medicine, University of Salamanca, 37007 Salamanca, Spain; 6Department of Physiology, Ribeirão Preto Medical School, University of São Paulo, Ribeirão Preto 14040-900, SP, Brazil; gabriel.smenezes@usp.br (G.S.-M.); ngcairas@usp.br (N.G.-C.); 7Bioinformatics, Intelligent Systems and Educational Technology (BISITE) Research Group, 37007 Salamanca, Spain

**Keywords:** animal models of epilepsy, audiogenic epilepsy, experimental surgery, vagus nerve stimulation

## Abstract

Vagus nerve stimulation (VNS) is an adjuvant neuromodulation therapy for the treatment of refractory epilepsy. However, the mechanisms behind its effectiveness are not fully understood. Our aim was to develop a VNS protocol for the Genetic Audiogenic Seizure Hamster from Salamanca (GASH/Sal) in order to evaluate the mechanisms of action of the therapy. The rodents were subject to VNS for 14 days using clinical stimulation parameters by implanting a clinically available neurostimulation device or our own prototype for laboratory animals. The neuroethological assessment of seizures and general behavior were performed before surgery, and after 7, 10, and 14 days of VNS. Moreover, potential side effects were examined. Finally, the expression of 23 inflammatory markers in plasma and the left-brain hemisphere was evaluated. VNS significantly reduced seizure severity in GASH/Sal without side effects. No differences were observed between the neurostimulation devices. GASH/Sal treated with VNS showed statistically significant reduced levels of interleukin IL-1β, monocyte chemoattractant protein MCP-1, matrix metalloproteinases (MMP-2, MMP-3), and tumor necrosis factor TNF-α in the brain. The described experimental design allows for the study of VNS effects and mechanisms of action using an implantable device. This was achieved in a model of convulsive seizures in which VNS is effective and shows an anti-inflammatory effect.

## 1. Introduction

Epilepsy shows a prevalence of 0.4–1% worldwide [1,2]. Up to a third of patients suffer from refractory epilepsy, in which seizure is not averted using two or more antiepileptic drugs [3]. Between 20 and 40% of these patients are candidates for surgical resection or disconnection procedures to disrupt seizure propagation [4]. Other alternatives include vagus nerve stimulation (VNS) [5], which consists of intermittent and chronic electrical stimulation of the left cervical vagus nerve [6].

The effectiveness of VNS peaks after six months of treatment, with a reduction of more than 50% in seizures in 45–65% of patients [7,8]. In addition, patients record an improvement in their quality-of-life metrics and comorbidities [9,10]. Moreover, VNS has proven to be safe, as VNS side effects are usually mild [11]. Furthermore, this treatment has a favorable cost–benefit ratio [12].

In the 1980s, Zabara revealed that stimulating the vagus nerve reduced seizures in dogs [13,14], and the first VNS surgery on a patient was performed [15]. In the 1990s, it was approved for treating refractory partial-onset epilepsy in adults [16]. Subsequently, its indication has been expanded to generalized and childhood epilepsies [17,18,19], as well as other complaints, including drug-resistant depression [20,21]. Moreover, new devices that include seizure detection algorithms based on cardiac function [22] and non-invasive approaches [23] have emerged.

However, despite its application to more than 125,000 patients [24], there is still debate about VNS mechanisms of action. The nucleus of the solitary tract receives a major synaptic input from the vagus nerve, and it is connected to several brain areas, including the reticular formation, the locus coeruleus, the parabrachial nucleus, the hypothalamus, and the amygdala [25]. Different hypotheses have been postulated, including that VNS causes changes in blood flow [26,27] or the desynchronization of the neuronal activity [28], the reduction in interictal epileptiform activity [29], and its effect on neurotransmitters [30]. In addition, an inflammatory reflex mediated by the vagus nerve has been reported [31], and VNS is being studied for inflammatory and autoimmune diseases [32,33]. Given the relationship between inflammation and epilepsy [34], an anti-inflammatory mechanism could also be partly responsible for the anticonvulsant effect of VNS [35].

VNS is a last resort, so clinical studies report heterogeneity, given that patients record a range of times of evolution, prior treatments, etiology, and types of epilepsy. It is within this context that animal models become crucial. However, although studies on animals have been important for the development of VNS therapy [14,36,37], most animal models do not allow to extrapolate their conclusions to clinical practice. This is due to the difficulty in both designing VNS systems for laboratory animals and developing a suitable experimental model that recreates clinical settings. Many animal studies are performed acutely, with the animal anesthetized or immobile, or applying arbitrarily selected stimulation parameters [38].

The Genetic Audiogenic Seizure Hamster from Salamanca (GASH/Sal) is a genetic model of audiogenic reflex epilepsy that shows generalized tonic–clonic seizures when exposed to intense sound [39]. Recent research has reported several approaches to the characterization of the model [40,41,42,43,44,45]. The current study describes an experimental model of VNS in the GASH/Sal that mimics clinical settings, with the aim being to delve further into the therapy’s mechanisms of action.

## 2. Results

### 2.1. Effects of Treatment on Seizures

Before surgery, the hamsters were subjected to acoustic stimulation to evaluate seizure severity by direct observation of the animal’s behavior. The rodents manifested the typical features of the seizures of the GASH/Sal, with behavioral arrest followed by wild running, tonic–clonic seizures, and post-ictal stupor. The mean value of the categorized seizure severity index (cSI) was approximately seven for all the groups (7.0 ± 1.83 for the SHAM group; 6.9 ± 1.97 for the animals in VNS groups). Therefore, seizure severity was considered high, since cSI scores go from 0 (absence of any seizure behavior) to 8 (tonic–clonic seizures, head ventral flexion, and limb extension).

Initially, GASH/Sal treated with VNS were divided in two groups: the first one using a clinically available neurostimulation device, and a second group in which the neuromodulation device used was specifically designed for small laboratory animals. Both groups that received VNS were analyzed together for statistical purposes because, as shown in Figure 1, the two neuromodulation devices proved to have similar effects in reducing seizure severity in the GASH/Sal, prompting the total absence of tonic–clonic seizures at the end of the experiment (Figure 1A,B). VNS therapy had a clear anticonvulsant effect in GASH/Sal, with a statistically significant reduction in seizure severity (*p* < 0.001). In addition, it is significant that the effectiveness of the treatment improved over time, as after seven days of treatment, there was a higher percentage of rodents that seized, with higher seizure scores than after ten and fourteen days of treatment (*p* < 0.05). However, there were no statistically significant differences between these last two records regarding seizure severity. Moreover, hamsters from the SHAM group did not display a reduction in seizure severity.

The effect of VNS on a subgroup of rodents was not only limited to the absence of tonic–clonic seizures, as there was also a total eradication of any epileptogenic behavior, including the wild-running phase, as shown in Figure 2. No statistical differences were observed between SHAM and VNS groups in the onset latency of the wild running phase.

Figure 3 shows the results of the neuroethological study (behavioral sequences). The pre-treatment records (Figure 3A) illustrate typical audiogenic seizures (sound-triggered) in GASH/Sal [44,45] (Appendix A) that are similar to other rodent strains [46,47,48]. In the pre-stimulation phase, it was possible to clearly see two behavioral clusters of exploration (SN, ER, WA, SC, IM) and of grooming (LI, GRG, GRF, GRH). The sound phase was initiated with startle (STA) followed by exploration behaviors and the replacement of the grooming behaviors with a cluster of so-called wild running to both sides, with the additional jumping an atonic falling (RU, GL, GR, JP e AF), colored in yellow. When the animals presented the tonic seizure (TCV), the sound was stopped, giving place to the post-stimulation phase. The latter was generally followed by strong seizure behaviors which characterize the generalized tonic–clonic seizures phase (TCV, HFL, HP_1_, HP_2_, CCV_1_, CVR_2_, CVL_2_ e CCV_1_), illustrated in red. The observed behavioral sequence was tonic–clonic seizures, followed by head ventral flexion and of the forelimbs and extension of the hindlimbs and clonic seizures. Those behaviors were followed by a stupor period (orange), or post-ictal immobility (PIM), where we were able to see respiratory alterations (TCP, DYS) or orofacial automatisms, such as eye blinking and masticatory movements (EB, MT), shown in purple. Figure 3B shows the neuroethological results of the SHAM group at the end of the experiment, evidencing no changes in seizures in the experimental group.

Except in the first test, in which all the GASH/Sal manifested epileptogenic behaviors, the rest of the analyses revealed a separation based on whether wild running was observed for each specimen (cSI ≠ 0). Figure 3C,E,G shows the data of the animals that did not display any abnormal behavior after 7, 10, and 14 days of treatment, respectively (cSI = 0). Startle on exposure to sound occurred, but this did not trigger, or was followed by, wild running behaviors, stupor, or tonic–clonic seizures (Appendix A).

Figure 3D,F,H contains the neuroethological evaluation of the animals treated with VNS that showed wild running (cSI ≠ 0) (Appendix A). In fact, during the 14 days of the VNS protocol, we were able to observe in the proportional time of the GL and GR behaviors of the wild running, a clear-cut cluster. The absence of tonic seizures or their presence in a few animals determined the exposure of the animals to a longer duration of the acoustic stimulation (maximum 60 s). Consequently, this situation determined a gradual reduction of the tonic–clonic seizure component. At the 10th day of the treatment, there was an accentuated reduction of seizure behaviors, with no interactions, displayed only by a single animal (Figure 3F). After 14 days of the VNS protocol, there was a complete absence of seizures (Figure 3H).

### 2.2. Effects of VNS on Behavior

Compared to the initial data in the open-field test, the GASH/Sal in both experimental groups showed a shorter distance travelled and time rearing, but longer grooming time (Figure 4). No statistical differences were detected between groups for any of the examined parameters. Note that the changes in the studied parameters seemed similar in both experimental groups. Therefore, surgery and the repetition of the test could have had the most far-reaching impact on the test results, albeit with no effect from the electrical stimulation of the vagus nerve.

### 2.3. Evaluation of Overall State

There was a reduction in body weight following surgery (Figure 5A). This reduction was slightly more pronounced in the rodents implanted with the device of clinical use, probably due to its high size compared to the animal. However, the procedure was thought to be well tolerated, and no adverse side effects were detected.

Regarding the hematological and biochemical parameters, no statistically significant difference was detected between the groups, nor any of the obtained abnormal data compared to previously published findings from our research group (Figure 5B) [49].

### 2.4. Electrophysiological Recordings

The electrophysiological recordings confirmed the effect that the ON phase of stimulation has on the vagus nerve (Figure 6).

### 2.5. Effect on Inflammatory Markers

Sixty minutes after the last evaluation of seizures, an anti-inflammatory effect was observed in the central nervous system, with a statistically significant reduction of the proinflammatory markers interleukin IL-1β, monocyte chemoattractant protein 1 (MCP-1), matrix metalloproteinase 2 (MMP-2), tumor necrosis factor alpha (TNF-α), and especially matrix metalloproteinase 3 (MMP-3) (Figure 7). However, a similar effect was not observed in plasma (Figure 8). In Figure 9, captures of a membrane from each group are shown, highlighting the markers for which differences were found in the left hemi brain samples.

## 3. Discussion

VNS causes an anticonvulsant effect in GASH/Sal, without producing any adverse side effects. The methodology used, with two different devices, provides similar and reproducible results, allowing this model to be used for studying the mechanisms of action of VNS. The behavioral evaluation of the audiogenic seizure phenotype, by means of neuroethological methods (behavioral sequences), is a strong measurement of the complexity of the current epileptic seizure model and of the anti-seizure effects of the VNS protocol. Not only it has been developed in audiogenic lines such as the Wistar Audiogenic Rat (WAR) strain [46], it has also been used in clinical settings [50,51,52].

From our perspective, the experimental model reported here has advantages over other models of VNS described in the literature [38,53]. First, it is a model that uses an implantable stimulation system [54], with the rodent moving about freely, thus removing the need for external stimulation and/or sedation. Second, it allows for the use of clinical parameters or the exploration of novel paradigms, as well as intermittent chronic stimulation over time, as performed clinically, not just for a few hours or minutes a day. In addition, it offers the possibility to conduct studies that last for months, given that it is estimated that the autonomy of the designed VNS device is of about a year, and the neurostimulation device, because it is small, is much more tolerable for the animal [54].

Moreover, the GASH/Sal has the advantage of being a genetic model that exhibits audiogenic reflex seizures (i.e., non-spontaneous) [39]. On the one hand, this could be an impediment given that the etiology of seizures is different from most human epilepsies [55], but on the other, it enables researchers to decide when the seizures are induced and when they are not, increasing the possibilities when proposing an experimental design and avoiding variability and interferences due to the seizures which may occur at any time [56]. In addition, seizures are not caused by electrical or chemical stimuli, thus avoiding potential interference between their induction and the study of anticonvulsant treatments. Furthermore, the performance of electrophysiological recordings in the nerve and cortical or other brain areas using stereotactic surgery paves the way for studying the electrophysiological effects of VNS on neuronal activity in the model.

Nonetheless, our experimental model has certain limitations. Although the hamsters’ response to the treatment has been outstanding, it has not been possible to obtain a cSI = 0 in all cases, so variations in the stimulation parameters could be explored to finally obtain a total absence of any abnormal behavior in all the animals after treatment. In turn, the clinical efficacy of VNS seems to improve over time [7], so we do not know whether the animals still exhibiting wild running after two weeks of treatment would continue to do so if the experiment were prolonged. The experiment was initially limited to 14 days: on the one hand, for bioethical reasons due to the size of the initially used commercial neurostimulation device, and on the other, to allow for the use of smaller batteries during the development of our own device. In addition, although in this study only animals aged 2–4 months were used, some GASH/Sal record a reduction in seizure severity after six months of age [39], which could be a problem for longer protocols. However, these hypotheses could be tested in future experiments.

Despite being a secondary matter in the research, open-field studies were carried out with the aim of supporting the reliability and safety of the procedure and the equipment used. Conducting these tests also shows that the VNS methodology applied in our work does not induce modifications in basic behaviors in animals, such as exploration and wandering. Additionally, according to our experience, we ponder that the high standard deviation values found after performing this test could have been due to individual differences exhibited between animals. In this context, we must also consider the fact that the behavioral patterns of hamsters are simpler and more repetitive than those of other rodents, such as the rat, for example. Although this was not a goal in our study, certainly the current VNS protocols can serve also to evaluate neuropsychiatric comorbidities and others, associated to the epilepsies we are modeling, such as anxiety, depression, and pain alterations, known to be present in experimental models such as the genetically selected audiogenic strains [57,58,59].

Regarding the study of inflammatory markers, from a methodological point of view, protein detection has been adequate, although the expression of some of the biomarkers has been very low compared to the positive control (biotin-conjugated IgG). This is probably due to its low expression under baseline conditions, or maybe because the antibodies used were optimized for mice, given the lack of protein arrays specifically designed for hamsters.

Alterations in the levels of inflammatory markers in blood, cerebrospinal fluid, and nervous tissue have been described in patients with epilepsy, and it has been postulated that these alterations could be involved in the pathophysiology of epilepsy [60,61]. However, the studies carried out in patients with epilepsy treated with VNS are very heterogeneous, since factors such as the etiology of epilepsy or pharmacological treatment can directly affect the levels of these markers. Among these studies, the research of Majoie et al. [62] stands out, describing a decrease in IL-6 in peripheral blood levels and an increase in levels of the anti-inflammatory cytokine IL-10 in responders to VNS.

Our results show the anti-inflammatory effect of this therapy on nervous tissue. Thus, a reduction in two of the most studied inflammatory markers, IL-1β and TNF-α, was observed. Both cytokines participate in mechanisms of epileptogenesis by promoting mechanisms of neuronal hyperexcitability and excitotoxicity. On the one hand, IL-1β is thought to induce seizures through upregulation of NMDA receptors [63]. In addition, intracerebral administration of IL-1β exacerbates seizures in rodent models, and seizures induce increased expression of this interleukin [64]. Moreover, it has been related to the genesis of febrile seizures [65], to the inhibition of neuronal plasticity mechanisms [66], and to the disruption of serotonergic transmission [67]. On the other hand, TNF-α induces the increase in AMPA receptors, stimulates the release of glutamate by microglia, and induces GABA receptor endocytosis [68]. In relation to VNS, the activity of the vagus nerve is known to be modified by exposure to proinflammatory cytokines, and particularly IL-1β [69]. According to Hosoi et al. [70], the activation of vagal afferents would produce an increase in the expression of IL-1β in the brain, thus activating the hypothalamic–pituitary–adrenal axis, to finally produce a systemic anti-inflammatory effect. This is an apparent contradiction with the observed results. However, in the mentioned study, the vagus nerve was cut, VNS lasted only 2 h, and different stimulation parameters were used. In addition, the measurement of interleukin levels was limited to hippocampus and hypothalamus, and the animals did not show seizures.

Moreover, a statistically significant reduction in the expression of MCP-1, also known as CCL2, was also detected. MCP-1 is a chemokine whose expression is increased in surgically resected brain tissue of patients with refractory epilepsy [71,72] and in animal models [73,74]. In animal studies, this increase has been demonstrated to be related to the recruitment of immune cells and neuronal damage [75,76].

Finally, a decrease in the levels of the extracellular matrix metalloproteinases MMP-2 and MMP-3 was observed. Experimental evidence suggests that MMPs are involved in seizure-induced cell death, breakdown of the blood–brain barrier, neuroinflammation, and aberrant synaptic plasticity, all of which occur in the context of epileptogenesis [77]. The levels of MMPs are increased in the brain of epileptic patients and in animal models of epilepsy [78,79]. The most studied MMP in epilepsy is MMP-9 [80], and it would be interesting to measure its levels in our model after VNS. We postulate that it would be similarly diminished in GASH/Sal treated with VNS. Although no studies that have evaluated MMPs using VNS in epilepsy have been found, non-invasive VNS reduces the levels of MMPs in the brain in rodent models of stroke [81]. Therefore, our results support the available literature.

However, a statistically significant anti-inflammatory effect in blood was not detected. This may have been due to several reasons derived from our experimental design and does not necessarily imply that VNS could not have similar anti-inflammatory effects at the systemic level. Firstly, seizures affect the nervous system, and the polarity of VNS implies a higher activation of the afferent vagus. Therefore, it seems logical that the greatest anti-inflammatory effect would be observed in brain tissue. Secondly, the half-life of some of these markers in the blood is of a few minutes [82], and perhaps there are changes in some of these molecules in blood in shorter periods of time after the induction of seizures, but after an hour, these levels may have already normalized to baseline conditions. It could also be possible that, in the event of an aggression of greater magnitude than an epileptic seizure, or in the case of another type of damage such as an infection, anti-inflammatory effects could be observed. Finally, the stimulation parameters used could also have impact on the systemic anti-inflammatory effect, since for the treatment of inflammatory diseases, a lower frequency of 10Hz is usually applied [83,84].

The experimental model described here allows for the study of VNS using an implantable stimulation system, which caters for the evaluation of stimulation parameters used clinically. Treatment with VNS has an anticonvulsant effect and reduces seizure severity in GASH/Sal. This effect is also progressive, increasing over time. In addition, it is a safe procedure, as no side effects were detected. Furthermore, our research supports the hypothesis of the anti-inflammatory effect of VNS. Finally, the design of an implantable neurostimulator adapted to rodents for basic research eliminates the need for external stimulation and renders it possible to conduct chronic studies, as well as reduce the costs of clinical neurostimulation devices.

In future studies, our research group will seek to delve further into the mechanisms of action underlying this therapy using the proposed model, aiming to contribute to our general understanding of epilepsy, in particular of this treatment.

## 4. Materials and Methods

### 4.1. Animals

A total of 30 GASH/Sal provided by the Animal’s Facilities at the University of Salamanca were used in this study. All of them were males and aged 2–4 months. Each one of the different protocols was performed at the same time of the day to avoid the influence of the circadian rhythm.

### 4.2. Ethics Statement

All the procedures involving animals and their care were conducted in accordance with the guidelines for the use and care of laboratory animals outlined in the Directive 2010/63/EU of the European Parliament and of the Council, in the current Spanish legislation (Royal Decree 1201/05), and in accordance with those established by the Institutional Bioethics Committee (approval number 942). All efforts were made to minimize animal suffering and reduce the number of rodents used.

### 4.3. Experimental Desgin

The hamsters (n = 28) were divided into three groups: the first one involved GASH/Sal treated with chronic VNS using a clinically available neurostimulation device (Group 1, n = 13). The second group involved GASH/Sal that received the same treatment, but the neuromodulation device used was specifically designed for small laboratory animals [54] (Group 2, n = 8). Finally, the VNS system designed was implanted in a third set of GASH/Sal, but it was switched off (SHAM group, n = 7).

First, the animals were subjected to an open-field test as an initial assessment of their behavior. Second, audiogenic seizures were recorded before surgery. The open-field test was repeated after 7 and 14 days of treatment, before acoustic stimulation and seizure evaluation, which were also performed on day 10 of the experiment. The GASH/Sal were weighed throughout the experiment, and hemogram and biochemical analyses were conducted at the end of the procedure, on seven animals treated with VNS and on four animals in the SHAM group. The study of inflammatory markers was performed in six animals of the VNS group and in five SHAM GASH/Sal, both in the blood and brain. In addition, the electrophysiological activity of the vagus nerve was recorded in two GASH/Sal to verify that the functionality of the vagus nerve was not affected. Euthanasia and sample collection were performed sixty minutes after the last exposure to sound. At the end of each experiment, the correct placing of the electrode was verified visually.

### 4.4. VNS Protocol

Both commercially available VNS devices and the research team’s own pulse generators were used in this study. On the one hand, this involved LivaNova ceded pulse generators (Model 104 VNS Therapy Demipulse Duo Generator, Cyberonics, Houston, TX, USA); a programming wand (VNS Therapy v8.1 software, DELL AXIM X50 Programming Wand Model 201 Cyberonics, Houston, TX, USA) was used to control battery levels, impedance values, and the system’s overall operation. Additionally, we used our own neuromodulation device, whose features have already been tested for safety, precision, and reproducibility [48].

The following stimulation parameters were used: 1.5 mA; 30 Hz; 250 µs pulse width, cycles 30 s ON/5 min OFF. Rodents received chronic treatment with these parameters for the whole duration of the experiment once the device was implanted. Given the size of the vagus nerve in hamsters, a bipolar electrode was designed for this purpose (Microprobes for Life Science, Maryland, USA), which consisted of two rings with an inner diameter of 500 µm, each one with an individual contact made of 50 µm platinum/iridium wire. The leads were 70 mm long with insulation.

### 4.5. Surgical Procedure

Surgery was performed using inhalation anesthesia under the surgical microscope, as described previously [54]. The left cervical vagus nerve was dissected and a sterile micropatch was placed around it. The bipolar electrode was then tunneled into the surgical area and placed around the nerve. Finally, the electrode was attached to the nerve by knotting a suture around its case, and the neuromodulation device was placed subcutaneously in the specimen’s dorsum. After surgery, the hamsters underwent a recovery period with water and food ad libitum in individual cages, receiving buprenorphine subcutaneously (0.5 mg/kg) as postoperative analgesia during the first three days.

### 4.6. Electrophysiological Recordings

Electrode implantation and electrophysiological recording were performed on the left vagus nerve of two GASH/Sal to demonstrate that the vagus nerve was not damaged by the implantation of the electrode. The specimen was first implanted with the VNS system as previously described. An additional stainless-steel wire-recording electrode was implanted in the vagus nerve cranial to the stimulation electrode. Next, an incision was made in the animal’s scalp, and the vagus’ recording electrode was embedded under the skin until it reached the incision. The animal was then positioned in a stereotaxic frame (David Kopf Instruments, Los Angeles, CA, USA). A reference (ground) stainless-steel screw electrode (epidural electrode) was implanted over the right frontal bone, and two additional screws were inserted into the skull to serve as anchors. All the electrodes were soldered to a connector, and all the wires were covered with dental acrylic (Henry Schein, Queens, NY, USA) to create a cap, closing the incision, and holding the connector. For the electrophysiological recording, the animal was placed inside a cylindrical acrylic chamber covered in a grounded stainless-steel mesh, thus acting as a Faraday cage. The head was connected to a tethered head stage preamplifier, connected to a multi-channel slip ring at the top of the chamber, allowing the specimen’s unrestricted movement without snagging the cable. Signal amplification and digitalization were performed at a 10 kHz sampling rate on PowerLab 4/20 and LabChart 7 v3.8 software (AD Instruments, Dunedin, New Zealand). Matlab R2022b (MathWorks, Natick, MA, USA) was used to preprocess and analyze the electrophysiological recording. The raw signal was filtered using a 5 Hz high-pass digital filter to attenuate low-frequency, high-amplitude oscillations in order to allow for better visualization of the stimulus.

### 4.7. Acoustic Stimuli

The procedure was performed according to the laboratory protocol [39]. The rodents were placed in a 37 cm diameter cylindrical arena and allowed to roam for one minute. They were then exposed to white noise (0–18 kHz; 115–120 dB) until they had tonic–clonic seizures, or a minute had elapsed. This procedure was video recorded for further neuroethological evaluation (see below). This test was performed before surgery and after 7, 10, and 14 days of treatment.

### 4.8. Neuroethological Analysis

Video recordings of the acoustic stimulation of GASH/Sal were analyzed using Ethomatic software at three time windows, namely, before, during, and after sound exposure [46,47]. The post-sound analysis lasted three minutes in the hamsters whose behavior involved at least wild running, whereas the evaluation for those that did not manifest any abnormal behavior was limited to one minute after the end of their exposure to sound. Flowcharts were drawn using Microsoft PowerPoint 365, where the animals with no epileptogenic activity are shown separately. Behavioral sequences were characterized as dyadic interactions (pairs of behaviors) described second by second as a probabilistic chain of events (see Figure 10B below).

The anticonvulsant effect of the therapy was evaluated using the categorized seizure severity index (cSI) [54]. In addition, the latency of the wild-running phase was also noted.

The cSI, the main features of Ethomatic software [46], and the dictionary of behaviors used are described in Figure 10.

### 4.9. Open-Field Tests

This procedure was carried out before and during the treatment with VNS (days 7 and 14). GASH/Sal were placed in the center of an 80 cm diameter arena rounded by a 30 cm high wall, which was divided into three concentric zones, each representing one-third of the surface. The tests lasted 12 min and were videorecorded from above. ANY-MAZE software (Stoelting Co., Wood Dale, IL, USA, v 6.16) was used for data collection: distance travelled, time spent in the central zone, and both grooming and rearing duration [52].

### 4.10. Hematological and Biochemical Liver Profiles

Blood samples for hemogram analysis and evaluation of inflammatory markers were obtained by cardiac puncture during euthanasia, transferred to EDTA-containing tubes, and processed in an ADVIA 120 cytometer (Bayer, Leverkusen, Germany). The parameters studied were hemoglobin, hematocrit, white blood cells, red blood cells, and platelet count.

In addition, the concentrations of total protein, bilirubin, albumin, aspartate aminotransferase, and alanine aminotransferase were measured in serum using standard laboratory kits (Spotchem II Liver-1 kit, #33925, Menarini Diagnostic, Badalona, Spain) in an automated Spotchem EZ analyzer (SP-4430). Accordingly, 400 μL of blood was extracted without anticoagulant substances, and 30 min later, the samples were centrifuged for 10 min at 10,000 rpm.

### 4.11. Study of Inflammatory Markers

The expression of 23 inflammatory markers in plasma and in the left hemi brain samples was evaluated using a commercial kit (Mouse Neuro Antibody Array ab211069, Abcam, Cambridge, UK). The proteins measured were Fas ligand (FasL); fractalkine; granulocyte colony-stimulating factor (G-CSF); interferon-gamma (IFN-γ); insulin-like growth factor 1 (IGF-1); interleukins IL-10, IL-1α, IL-1β, IL-4, and IL-6; keratinocyte-derived cytokine (KC); LPS-induced chemokine (LIX); MCP-1; macrophage-colony-stimulating factor (M-CSF); macrophage inflammatory protein 1 alpha (MIP-1α); MMP-2; MMP-3; receptor for advanced glycation endproducts (RAGE); stromal-cell-derived factor 1 alpha (SDF-1α); thymus-activation-regulated chemokine (TARC); transforming growth factor beta (TGF-β); TNF-α; and vascular endothelial growth factor A (VEGF-A), following the manufacturer’s instructions.

On the one hand, to extract protein from the whole left-brain, samples were weighted, and a mix containing the lysis buffer from the commercial kit and protease inhibitors was added (Protease/phosphatase inhibitor cocktail #5872, Cell Signaling Technology, Danvers, MA, USA; PMSF 36978, Thermo Fisher Scientific, Waltham, MA, USA) according to the ratio of 1ml of buffer per 100mg of tissue. The samples were mechanically homogenized (IKA T10 basic Ultra Turrax Homogenizer Workcenter, IKA, Staufen, Germany). Once centrifuged at 14,000 rpm and 4 °C for 15 min, the supernatant was collected and stored until use at −70 °C. On the other hand, plasma was obtained from the centrifugation of blood samples (see above) at 4500 rpm for 10 min.

The manufacturer’s instructions were followed. A 1:8 dilution factor was applied to the left hemi brain samples to reduce protein concentration and background. Incubation steps were carried out overnight at 4 °C. Exposure time for chemiluminescence detection was two minutes. The densitometric analysis of the images was performed using ImageJ software v1.53k [85]. The signal obtained for each marker was normalized to the positive control spots (amount of biotin-conjugated IgG) as an indication of protein concentration. Then, the chemiluminescence obtained for each sample in the VNS group was normalized to the mean obtained for the SHAM group to facilitate data visualization.

### 4.12. Statistical Analysis

Data analysis was performed using IBM SPSS Statistics software, v.26 (SPSS Inc., USA). Statistical significance levels were *p* < 0.05 (*), *p* < 0.01 (**), and *p* < 0.001 (***). All data are shown as mean ± standard deviation (SD).

Seizure severity before and after treatment was evaluated by repeated-measures tests (ANOVA non-parametric repeated measures tests, Durbin–Conover for post hoc comparisons). The SHAM and VNS groups were compared using the Mann–Whitney U test because the data did not exhibit normality.

To elaborate flowcharts using Ethomatic, statistical analysis of the strength of a given behavioral sequence was performed with the χ^2^ test, whose log values were proportional to the width of the arrows shown. Behavior frequency was defined as the mean number of times of a given behavior in the observation period, whereas the duration of behavior was defined as the mean duration of a given behavior in the observation period.

Data variables were tested for normality to analyze open-field test results. Parametric or non-parametric tests were therefore used accordingly. One-way ANOVA or the Friedman test (Tukey and Durbin–Conover post hoc comparisons, respectively) was used to compare the results obtained before and during treatment for the same group. Student’s *t* or Mann–Whitney tests were performed to compare SHAM and VNS groups at the same time.

Finally, a comparison between levels of each inflammatory marker between groups was performed using unpaired Student’s *t* tests. 

## Figures and Tables

**Figure 1 ijms-25-00091-f001:**
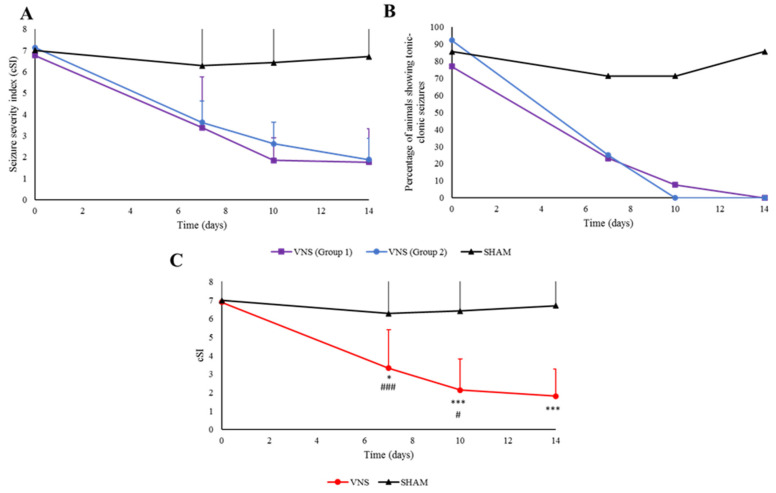
Anticonvulsant effect of VNS. (**A**) Comparative effectiveness of the neuromodulation devices used to reduce seizure severity in contrast with the SHAM group. (**B**) Number of GASH/Sal exhibiting tonic–clonic seizures. Notice that none of the animals treated with VNS had seizures at the end of the experiment. (**C**) Seizure severity of animals treated with VNS. GASH/Sal from groups 1 and 2 are shown together. There was a significant anticonvulsant effect in animals treated with VNS, which improved with time. Vertical bars indicate hemi standard deviations. Note that ‘*’ is used to indicate statistically significant differences between VNS and SHAM groups, whereas ‘#’ shows the differences in the VNS group when compared to the previous records. ‘*’ (*p* < 0.05); ‘***’ (*p* < 0.001). ‘#’ (*p* < 0.05); ‘###’ (*p* < 0.001).

**Figure 2 ijms-25-00091-f002:**
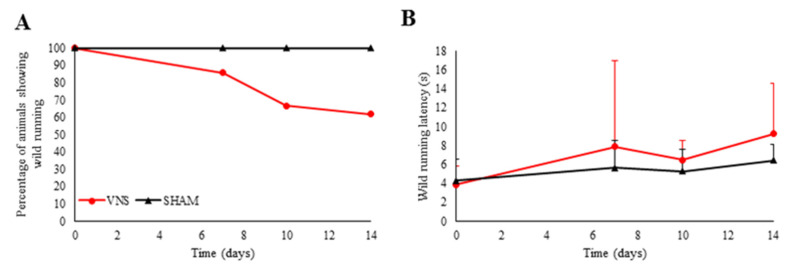
Effect of treatment on wild running. (**A**) Number of GASH/Sal exhibiting wild running. VNS reduced the number of hamsters exhibiting this behavior, which was observed in all the experimental animals before treatment and in all the subjects of the SHAM group. (**B**) Effect of treatment on the latency of wild running. VNS tends to increase latency in the GASH/Sal with this type of behavior. Vertical bars indicate hemi standard deviations.

**Figure 3 ijms-25-00091-f003:**
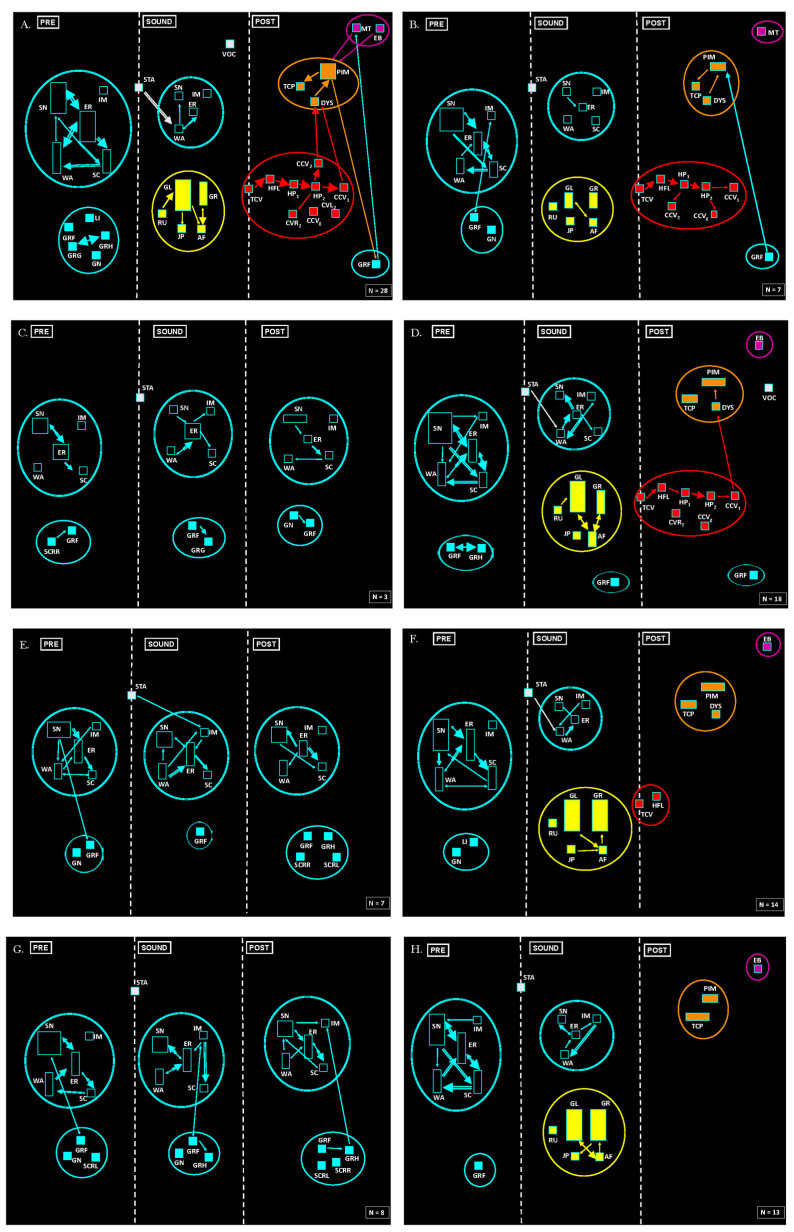
Effect of VNS on audiogenic seizures. Flowcharts showing the sequence of behaviors in different experimental groups at various times: (**A**) GASH/Sal in baseline conditions; VNS and SHAM groups evaluated together. (**B**) SHAM group at the end of the experiment. (**C**) VNS group on day 7 of treatment (cSI = 0). (**D**) VNS group on day 7 of treatment (cSI ≠ 0). (**E**) VNS group after 10 days of treatment (cSI = 0). (**F**) VNS group after 10 days of treatment (cSI ≠ 0). (**G**) VNS group on day 14 of treatment (cSI = 0). (**H**) VNS group at day 14 of treatment (cSI ≠ 0). Note that VNS eliminated behaviors associated with tonic–clonic seizures in GASH/Sal (shown in red circle), and some of them did not even display any behavioral arrest or wild running phase (in yellow circle).

**Figure 4 ijms-25-00091-f004:**
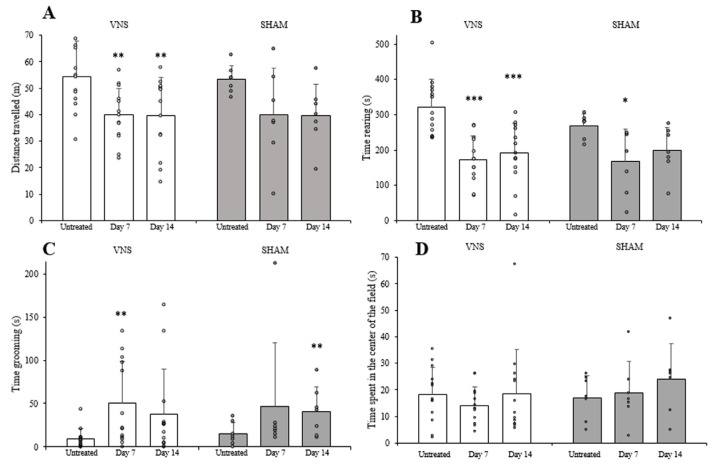
Effect of treatment on behavior. Open-field test scores obtained in distance travelled (**A**), time rearing (**B**), time spent in the center of the field (**C**), and grooming duration (**D**). ‘*’ indicates differences between repeated tests in each experimental group. No differences were detected between the VNS (white) and SHAM (gray) groups. Vertical bars indicate hemi standard deviations. Points represent single experiment results ‘*’ (*p* < 0.05); ‘**’ (*p* < 0.01); ‘***’ (*p* < 0.001).

**Figure 5 ijms-25-00091-f005:**
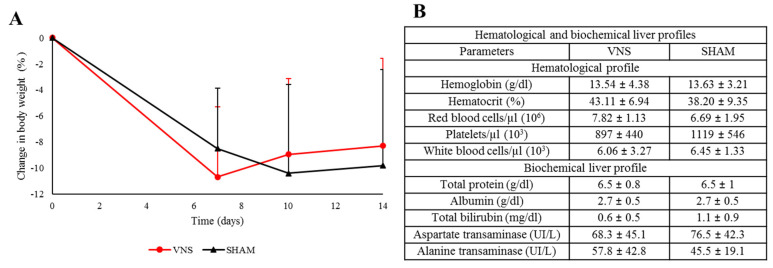
Effect on normofunction. (**A**) Body weight. A reduction in body weight was observed in both groups following surgery. (**B**) Hematological and biochemical liver profiles. No differences were detected between the groups. Values for all the measured parameters are considered non-pathological. Vertical bars indicate hemi standard deviations.

**Figure 6 ijms-25-00091-f006:**
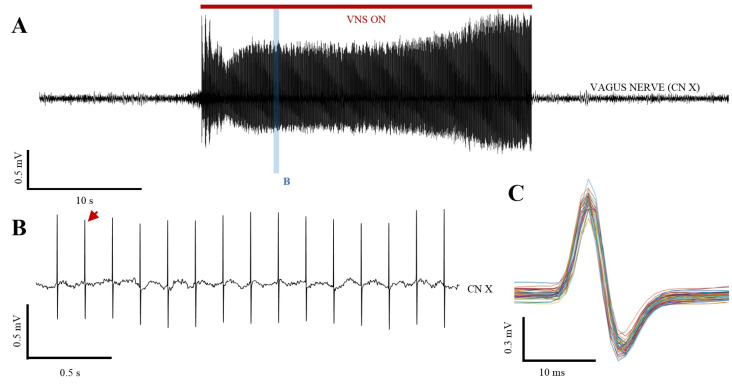
Electrophysiological recordings of the vagus nerve (CN X) during the VNS protocol. (**A**) Representative local field potential recordings before, during, and after the 30 s stimulation train. The red line represents the stimulus duration, and the blue band marks the recordings in B. (**B**) A short sample of the recordings in A showing the electrical stimulus in more detail. The red arrow points to the bipolar stimulation spikes, which can be seen in the CN X. (**C**) Close-up view of the shape of 50 sequential spikes showing the bipolar waveform of the stimulation.

**Figure 7 ijms-25-00091-f007:**
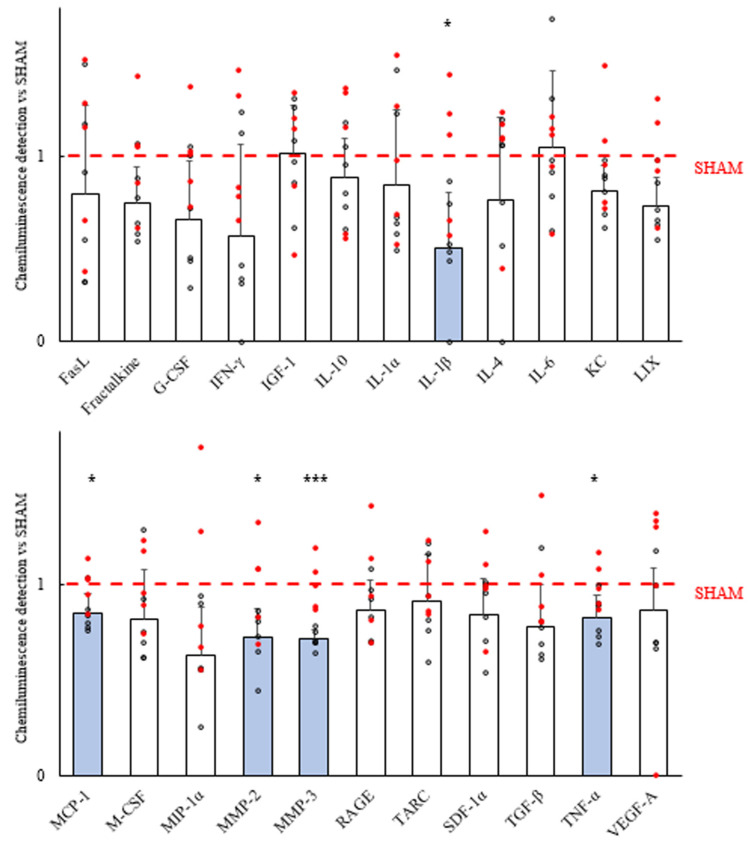
Brain inflammatory profiles of GASH/Sal treated with VNS. Bars represent the mean values obtained in the VNS group after being normalized to the SHAM group (in red) and the positive controls of each membrane. FasL: Fas ligand; G-CSF: granulocyte-colony-stimulating factor; IFN-γ: interferon-gamma; IGF-1: insulin-like growth factor 1; IL: interleukin; KC: keratinocyte-derived cytokine; LIX: LPS-induced chemokine; MCP-1: monocyte chemoattractant protein-1; M-CSF: macrophage-colony-stimulating factor; MIP-1α: macrophage inflammatory protein 1 alpha; MMP: matrix metalloproteinase; RAGE: receptor for advanced glycation end products; SDF-1α: stromal-cell-derived factor 1 alpha; TARC: thymus-activation-regulated chemokine; TGF-β: transforming growth factor beta; TNF-α: tumor necrosis factor alpha; VEGF-A: vascular endothelial growth factor A. The markers for which statistical differences were detected (IL-1β, MCP-1, MMP-2, MMP-3, TNF-α) are shown in blue. Dots represent individual results. Vertical bars indicate hemi standard deviations. ‘*’ (*p* < 0.05); ‘***’ (*p* < 0.001).

**Figure 8 ijms-25-00091-f008:**
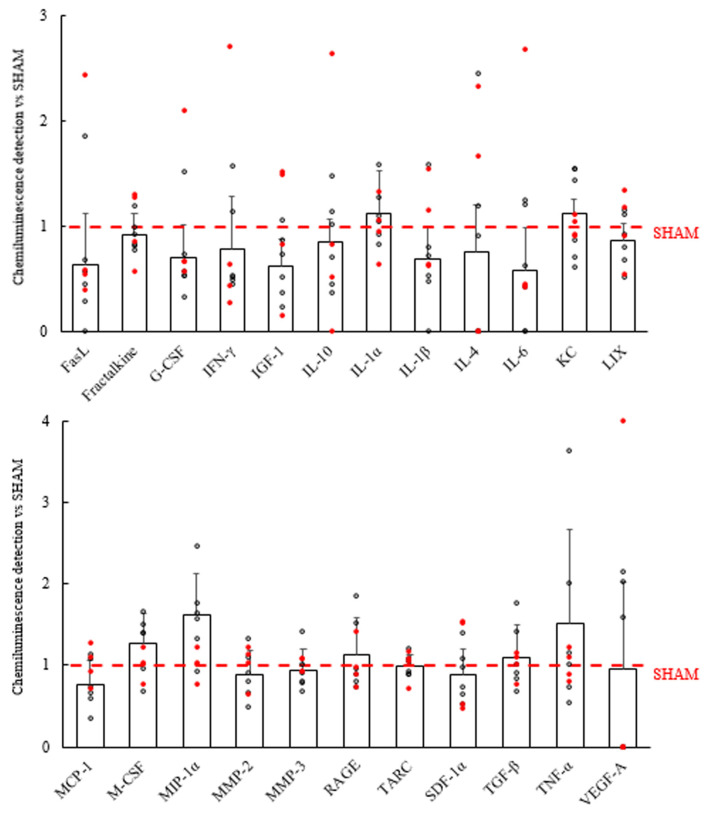
Plasma inflammatory profiles of GASH/Sal treated with VNS. Bars indicate the mean values obtained in the VNS group after being normalized to the SHAM group (in red) and the positive controls of each membrane. Dots represent individual results. Vertical bars indicate hemi standard deviations. No statistical differences were detected.

**Figure 9 ijms-25-00091-f009:**
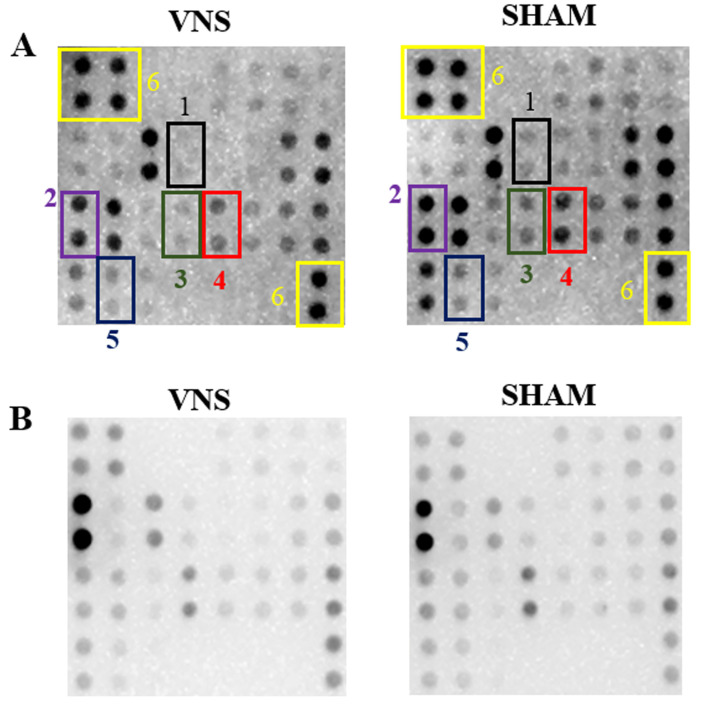
Captions of one array of each group and type of sample studied. (**A**) Example of the results obtained in the brain. Spots with numbers indicate the markers for which statistical differences were detected. (1) IL-1β (black). (2) MCP-1 (purple). (3) MMP-2 (green). (4) MMP-3 (red). (5) TNF-α (blue). (6) Positive control spots (yellow). (**B**) Caption of results obtained in plasma.

**Figure 10 ijms-25-00091-f010:**
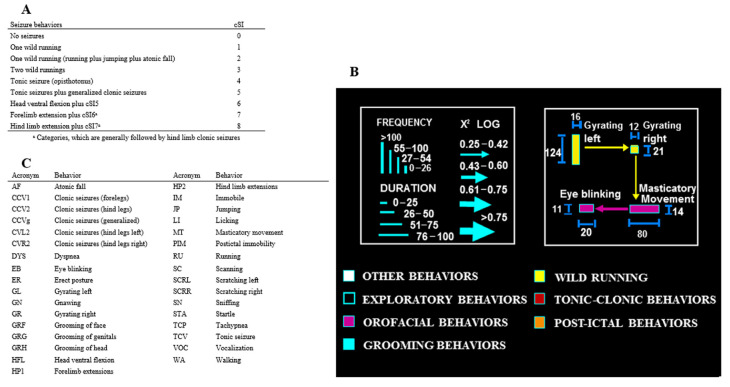
Methods for seizure assessment. (**A**) Description of cSI, which allows for the categorization of seizures on a scale from 0 to 8 based on behaviors observed for a statistical approach. (**B**) Ethomatic software function [46]. Ethomatic classifies the behaviors observed according to a dictionary of items and shows the frequency and duration of each of them. In addition, this allows for the performance of a statistical analysis of the probabilistic association between pairs (dyads) of items using the χ^2^ test. The frequency and time spent performing each behavior were proportional to the height and width of each rectangle, respectively. The width of each arrow between dyads was proportional to its χ2 log value and thus to statistical significance, whereas its direction indicated the preference association between the two items. The color of each rectangle was also associated with specific classes or clusters of behaviors. (**C**) Dictionary of behaviors used to perform the neuroethological studies.

## Data Availability

The data presented in this study are available on request from the corresponding author.

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
