# Peer review of "Effect of Vagus Nerve Stimulation on the GASH/Sal Audiogenic-Seizure-Prone Hamster"

_ijms, 2023, doi:10.3390/ijms25010091_

Round 1

Reviewer 1 Report

Comments and Suggestions for Authors

In this manuscript, the authors demonstrate the impacts of vagal nerve electrical stimulation on the behavioural and immune-histochemical features of epilepsy in a specific rodent genetic model of auditory-evoked seizures.  In particular, the author show that a home-made, miniature and implantable electric stimulator device can effectively reduce the severity of the evoked seizure symptoms.  The tonic-clonic seizures are ameliorated over the course of 1-2 weeks stimulation in most rodents and this is associated with a decrease in some brain inflammatory markers. The motor symptoms of running typically remain and whether the underlying EEG seizure activity is attenuated is not examined.  There are no obvious side-effects within the few parameters measured.

Vagal nerve stimulation is approved clinically for different refractory epilepsy syndromes and has been evaluated in different animal models. The precise mechanisms for its effect are unclear, warranty further animal studies. This manuscript provides another animal epilepsy model where VNS is evaluated, and furthermore validates a miniature stimulation device that may be of use for other researchers. The data is mostly clearly portrayed and the manuscript is clear. A few aspects are over-interpreted and some further clarifications requested.

1. (line 86) Define how the seizure severity index is measured. And not just as an output from the software.

2. (line 83 and throughout). The hamsters aren’t “specimens” that suggests some in vitro or tissue sample. Re-word as “rodents”, “hamsters” or groups etc.

3. (line 89) “Both groups” – the reader has not yet been introduced to the experiments where two different stimulation devices were used. I suggest a brief sentence to introduce the three different experimental groups either here or at start of Results paragraph 1.

4. (lines 94-96) “In addition, it is significant that the effectiveness of the..” Delete this. Saying “significant” implies the time course of effect was somehow evaluated. The cSI at 7, 10 and 14 days may be the same. Simply state something like:. “The effectiveness of the treatment may have improved over time, based on the lower % of rodents showing tonic-clonic seizures at (0%) as compared to 7 days (20-30%)”.

5. Lines 112-114. The latency to wild running (Fig 2B) is not different between Sham and test so don’t claim any “tendency” – just say wasn’t different.

6. Figure 3 is too complex for the reader with many abbreviations, samples and arrows not explained or very hard to follow. Find a different way to show this data (eg a timeline from stimulus that maps out different grouped behaviours). The description lines 121-142 also quite complex, but will also be rewritten with a revised and simplified Figure.

7. Lines 156-158. If no statistical difference then don’t claim any “higher increase for the SHAM”

8. Lines 161-162. “Statistical significance..” delete this sentence. The different level of significance cant be interpreted this way. As you say, may reflect sampling or other factors.

9. Figures 7-9. These intensity / densitometric graphs were normalised to some control protein (“positive control”, line 477). Was it cortex or whole left brain? Provide details in 4.11 how the brains were isolated and homogenized for the immunoblots, and how this was done.  What the control protein and please label it in Figure 9.

10. Was the VNS run continuously (30s On/5 min off) for the 14 days? State this detail please.

Reviewer 2 Report

Comments and Suggestions for Authors

The manuscript by Gonçalves-Sánchez et al. investigate the anticonvulsant effects of Vagus Nerve Stimulation (VNS) in a genetic model of epilepsy known as GASH/Sal. The GASH/Sal model, characterized by audiogenic reflex seizures, allows researchers to control when seizures are induced, reducing variability in experimental design. The GASH/Sal hamsters underwent VNS for 14 days using clinical stimulation parameters by implanting either a commercially available neurostimulation device or their own prototype for laboratory animals. The study concludes that VNS using an implantable stimulation system is effective in reducing seizure severity in the GASH/Sal model. It also supports the hypothesis of VNS having anti-inflammatory effects.

As pointed out in the discussion, further investigation can be done using the GASH/Sal model to analyze the effect of VNS over a longer period of time and to further characterize the molecular mechanisms of VNS in epilepsy. I find this manuscript an interesting preliminary study that, as outlined in the discussion, shows some limitations that the authors are aware of.

Here are some minor revisions:

  1. The authors state that the two neuromodulation devices proved to have similar effects in reducing seizure severity in the GASH/Sal and for this reason both groups of VNS were analyzed together. I was wondering if the absence of difference between the two neuromodulation devices applies also for the other test (behaviour test, Electrophysiological recordings…). Did you analyze the data from the two group (commercially available VNS and homemade VNS devices) separately?

  1. While not the primary focus of this paper, briefly highlighting the advantages of your VNS device in comparison to the commercial one could be intriguing for the reader. This is especially relevant considering both devices demonstrate a similar effectiveness in reducing seizure severity in the GASH/Sal.

  1. In the behavioral test (paragraph 2.2) the changes in the behavioral parameters follow the same pattern in both VNS and SHAM group. Moreover, the data are inconsistent and shows really high standard deviations. I was wondering what the relevance of these behavioral parameters is (distance travelled, rearing and grooming time) in the GASH/Sal model and in reference to VNS. It might be worthwhile to briefly elaborate on this in the paragraph 2.2 or in the discussion.

  1. For the figures with bar graph (Figure 4, 7 and 8) I suggest changing the type of graph and use a bar graph with also individual points (representing the single experiment/animals). With this type of representation, it will be clearer the distribution of your data.

Round 2

Reviewer 1 Report

Comments and Suggestions for Authors

Thanks for addressing my initial comments.

Reviewer 2 Report

Comments and Suggestions for Authors

The authors satisfied all the points of my review, so I have no further comments.